# Risk of Severe COVID-19-Related Outcomes among Patients with Cirrhosis: A Population-Based Cohort Study in Canada

**DOI:** 10.3390/v16030351

**Published:** 2024-02-24

**Authors:** Héctor Alexander Velásquez García, Prince A. Adu, Ada Okonkwo-Dappa, Jean Damascene Makuza, Georgine Cua, Mawuena Binka, James Wilton, Hind Sbihi, Naveed Z. Janjua

**Affiliations:** 1British Columbia Centre for Disease Control, Vancouver, BC V5Z 4R4, Canada; hector.velasquez@bccdc.ca (H.A.V.G.); jean.makuza@bccdc.ca (J.D.M.); georgine.cua@bccdc.ca (G.C.); james.wilton@bccdc.ca (J.W.); hind.sbihi@bccdc.ca (H.S.); 2School of Population and Public Health, University of British Columbia, Vancouver, BC V6T 1Z3, Canada; mawuena.binka@bccdc.ca; 3Department of Social Medicine, Heritage College of Osteopathic Medicine, Ohio University, Dublin, OH 43016, USA; 4Department of Family Practice, University of British Columbia, Vancouver, BC V6T 1Z4, Canada; ada.okonkwodappa@ubc.ca; 5Centre for Advancing Health Outcomes, St. Paul’s Hospital, Vancouver, BC V6Z 1Y6, Canada

**Keywords:** COVID-19, cirrhosis, risk, hospitalization, risk factors, Canada

## Abstract

We assessed the association between cirrhosis and severe COVID-19-related outcomes among people with laboratory-diagnosed COVID-19 infection in British Columbia, Canada. We used data from the British Columbia (BC) COVID-19 Cohort, a population-based cohort that integrates data on all individuals tested for COVID-19, with data on hospitalizations, medical visits, emergency room visits, prescription drugs, chronic conditions, and deaths in the Canadian province of BC. We included all individuals aged ≥18 who tested positive for SARS-CoV-2 by real-time reverse transcription-polymerase chain reaction from 1 January 2021 to 31 December 2021. Multivariable logistic regression models were used to assess the associations of cirrhosis status with COVID-19-related hospitalization and with ICU admission. Of the 162,509 individuals who tested positive for SARS-CoV-2 and were included in the analysis, 768 (0.5%) had cirrhosis. In the multivariable models, cirrhosis was associated with increased odds of hospitalization (aOR = 1.97, 95% CI: 1.58–2.47) and ICU admission (aOR = 3.33, 95% CI: 2.56–4.35). In the analyses stratified by age, we found that the increased odds of ICU admission among people with cirrhosis were present in all the assessed age-groups. Cirrhosis is associated with increased odds of hospitalization and ICU admission among COVID-19 patients.

## 1. Introduction

In 2020, the World Health Organization (WHO) declared the outbreak of coronavirus disease 2019 (COVID-19), an infectious disease caused by the severe acute respiratory syndrome coronavirus-2 (SARS-CoV-2), to be a pandemic. As of March 2023, COVID-19 has led to over six million recorded deaths worldwide [1]. In British Columbia (BC), Canada’s third largest province by population size, 34,961 people required hospitalization for COVID-19, and over 5430 deaths have been reported as of 15 April 2023 [2]. COVID-19 viral infection in healthier individuals is more likely to manifest as a mild-to-moderate respiratory illness. In contrast, individuals with pre-existing illnesses and comorbidities may experience more severe outcomes [3,4,5].

People with cirrhosis may be at higher risk of severe outcomes. A study examining the rates of 30-day mortality in patients with cirrhosis and COVID-19 found that 96% of people with cirrhosis required hospitalization [6]. The causes of cirrhosis vary geographically, with people with chronic hepatitis C and non-alcoholic fatty liver disease (NAFLD) being the most common cases in western countries [7] and chronic hepatitis B being the primary cause of liver cirrhosis in the Asia-Pacific region [8]. Generally, people with cirrhosis develop immune dysfunction, making them particularly exposed to an elevated risk of infections and their associated complications [9], and have poorer outcomes from acute respiratory distress syndrome than patients who do not have cirrhosis [10]. Higher rates of liver dysfunction have also been reported among patients with severe COVID-19 [11].

Given the immunocompromised status of patients with cirrhosis, a better understanding of their risk factors for COVID-19 severe outcomes is critical for treatment and preventive efforts [11]. While there exists literature addressing the risk of COVID-19 outcomes in individuals with cirrhosis, the majority of these studies exhibit limitations, particularly in terms of sample size; a significant proportion of investigations on the subject have been constrained by relatively smaller cohorts (usually hospital-based), potentially influencing the generalizability of their findings. Moreover, a notable gap exists in the representation of North American perspectives within this body of work. Only a limited number of population-based studies have specifically delved into the context of North America, leaving a noteworthy gap in our understanding of how COVID-19 impacts individuals with cirrhosis in this region. Therefore, the aim of this study was to address this important knowledge gap. We assessed the association of cirrhosis with severe COVID-19-related outcomes among people with laboratory-diagnosed COVID-19 cases in British Columbia (BC).

## 2. Materials and Methods

### 2.1. Study Design and Data Sources

We used data from the BC COVID-19 Cohort (BCC19C)—a population-based surveillance platform established under the BC Centre for Disease Control (BCCDC)’s public health mandate. This platform integrates data on all individuals tested for COVID-19 in BC with data on COVID-19 hospital and intensive care unit admissions, medical visits, all hospitalizations, emergency room visits, chronic conditions, prescription drugs, and mortality (See Appendix A).

### 2.2. Study Population

The analyses included all individuals aged 18 or above who tested positive for SARS-CoV-2 by real-time reverse transcription–polymerase chain reaction (RT-PCR) from 1 January 2021 to 31 December 2021. We excluded individuals who resided in long term care facilities, as hospital transfers for these individuals were irregular over time and across local regions. The risk profiles of these individuals were also different from the general population.

### 2.3. Outcome and Exposure

We assessed two main outcomes of interest: hospitalization and intensive care unit (ICU) admission. Hospitalization was defined as a hospital admission in a BC acute care facility within 14 days after a positive SARS-CoV-2 test. ICU admission was defined as being admitted to the ICU during hospitalization within 14 days after a positive SARS-CoV-2 test. We assessed hospitalization in two ways: one that excludes ICU admission, and another that includes ICU admission to assess the severity gradient.

We considered cirrhosis to be the primary exposure. We also assessed the following comorbidities and risk factors: Alzheimer/dementia, asthma, acute myocardial infarction, chronic obstructive pulmonary disease (COPD), chronic kidney disease (CKD), depression, diabetes (categorized as no-diabetes, treated without insulin, or requiring insulin), epilepsy, obesity, weight loss, parkinsonism, rheumatoid arthritis, injection drug use, alcohol misuse, cancer, immunosuppression, intellectual and developmental disabilities, schizophrenia and psychotic disorders, income (categorized as quintiles), COVID-19 vaccination status, SARS-CoV-2 variant of concern, age, sex, and regional health authority. The definitions and diagnostic codes used to identify the comorbidities are presented in Appendix A).

### 2.4. Statistical Analysis

We compared demographic characteristics and risk factors by COVID-19 severity (hospitalization and ICU admission status), and also by cirrhosis status. The associations of cirrhosis with hospitalization and with ICU admission were assessed by estimating odds ratios through multivariable logistic regression models. We then stratified these analyses by age-group.

We also repeated these analyses by including ICU cases in the hospitalization cases. For the models assessing ICU as outcome, we examined both ICU vs. non-hospitalization and ICU vs. hospitalization.

### 2.5. Ethical Approval

This study was reviewed and approved by the Research Ethics Board at the University of British Columbia (approval # H20-02097).

## 3. Results

### 3.1. Demographic Characteristics

Characteristics of the study population according to COVID-19 severity are presented in Table 1. Of the 162,509 individuals testing positive for SARS-CoV-2 who were included in the analysis, there were 6035 (94.7%) hospital admissions and 2511 (1.5%) ICU admissions. Overall, 768 (0.5%) had cirrhosis, 96,760 (59.5%) were not vaccinated, 17,199 (10.6%) were partially vaccinated, and 48,550 (29.9%) were fully vaccinated.

Among those who were hospitalized, 132 (2.2%) had cirrhosis, compared to 98 (3.9%) among those who were admitted to the ICU and 538 (0.3%) among those who were not hospitalized. The median ages were 38 years (IQR: 28–52) for all COVID-19 cases, 60 years (IQR: 44–73) for hospital admission, 60 years (IQR: 48–70) for individuals who were admitted to the ICU, and 37 years (IQR: 27–50) for individuals who had no hospitalization. A lower proportion of hospital admission cases (54.8%) were males, compared to the males who were admitted to the ICU (63.0%). Also, a greater proportion of hospital admission cases (30.4%) were in the lowest income quintile, compared to those admitted to the ICU (29.7%) and those with no hospitalization (18.8%).

A higher proportion of individuals who were admitted to the ICU had comorbidities, compared to those who either experienced only hospital admission or no hospitalization. For example, asthma (18.5% vs. 18.3% vs. 13.3%), lymphoma (2.3% vs. 1.9% vs. 0.5%), metastatic cancer (5.5% vs. 5.3 vs. 1.9%), chronic kidney disease (19.8% vs. 18.3% vs. 2.7%), COPD (10.6% vs. 10.1% vs. 1.4%), insulin-dependent diabetes (9.8% vs. 6.6% vs. 1.1%), non-insulin-dependent diabetes (18.4% vs. 17.0% vs. 5.1%), obesity (8.4% vs. 4.5% vs. 2.7%), hypertension (40.5% vs. 40.0% vs. 11.3%), and immunosuppression (7.6% vs. 6.1% vs. 2.3%) [Table 1]. The proportion who were unvaccinated was highest among ICU admissions (82.4%), followed by hospital admission cases (73.6%), compared to the non-hospitalized population (58.6%) [Table 1].

Characteristics of the study population according to cirrhosis status are presented in Table 2. Among those diagnosed with cirrhosis, 17.2% were hospitalized, 12.8% required admission to an ICU, and 70.1% did not require hospitalization. Individuals with cirrhosis had a higher median age [54 years (43–64)], compared to those with no cirrhosis [38 years (28–52)]. There was a higher proportion of people in the lowest income quantile among individuals with cirrhosis (34.9%), compared to those without cirrhosis (19.4%). Also, for most of the comorbidities that were assessed, a higher proportion of them were present among individuals with cirrhosis, compared to those without cirrhosis [Table 2].

### 3.2. Risk Factors Overall

In the adjusted models, cirrhosis was associated with increased odds of hospitalization (adjusted odds ratio [aOR] = 1.97, 95% CI: 1.58, 2.47) and ICU admission (aOR = 3.33, 95% CI: 2.56, 4.35) [Table 3; Figure 1]. For the model in which ICU admissions was compared with hospitalization cases (instead of no-hospitalization cases), the magnitude of association was slightly attenuated, although still significant (aOR = 1.84, 95% CI: 1.38, 2.46). Also, in the model in which hospitalization cases included ICU cases, the magnitude of association increased (aOR = 2.55, 95% CI: 2.11, 3.08) [Appendix A; Figure 1].

### 3.3. Cirrhosis and COVID-19 Severity by Age-Group

The increased odds of hospitalization among individuals with cirrhosis were greatest among the youngest age-group (18–49 years) (aOR = 2.65; 95% CI: 1.77–3.99), followed by the 50–69 age-group (aOR = 2.04; 95% CI: 1.50–2.78), and then the ≥70 age-group (aOR = 1.48; 95% CI: 0.90–2.46) [Table 3; Figure 1]. When we included ICU cases in the hospitalization cases, a similar trend was observed, although the effect sizes were increased (Appendix A; Figure 1).

In a similar fashion, the increased odds of ICU admission among individuals with cirrhosis were greatest among the youngest age-group (18–49 years) (aOR = 6.81; 95% CI: 4.31–10.78), followed by the 50–69 age-group (aOR = 2.98; 95% CI: 2.08–4.27), and finally, the ≥70 age-group (aOR = 2.17; 95% CI: 1.13–4.17) [Table 3; Figure 1]. Although with attenuated effect sizes, a similar trend was observed when we made hospitalization cases the reference category, instead of no-hospitalization cases (Appendix A).

## 4. Discussion

Despite the strong evidence supporting the associations between various comorbidities and poor COVID-19 prognosis, population-level evidence for the association between cirrhosis and COVID-19 severe outcomes remains an important knowledge gap. This large, provincial, population-based cohort study assessed the risk of COVID-19 severe outcomes among people with cirrhosis using data from confirmed COVID-19 cases from 1 January 2021 to 31 December 2021 in BC, Canada. Findings from our analysis indicate that individuals with cirrhosis faced elevated risks of both hospitalization and admission to the ICU, in the context of COVID-19 infection. Similar findings have been observed in other studies. In Chile, Díaz and colleagues found increased hospitalization (42.9% vs. 7.7% in the overall population) among patients with COVID-19 and underlying cirrhosis [12]. Their figures compare to the 12.2% hospitalization rate (excluding ICU admission) recorded in our study. Our ability to assess cirrhosis as an independent risk factor addresses the limitation in the Chilean study. Specifically, our findings showed that among individuals who have tested positive for COVID-19, those with cirrhosis had 97% and 223% greater risk of hospitalization and ICU admission, respectively, compared to individuals without cirrhosis.

Our findings are also consistent with a large cohort study in the U.K. which found that individuals with pre-existing conditions, including cirrhosis of the liver, had an increased risk of COVID-19-related hospitalization or death [13]. Marjot and colleagues also found that patients with cirrhosis had an increased risk of death from COVID-19 [4]. More recently, a study in the United States found that COVID-19-positive veterans with cirrhosis had increased risks of severe disease and death, compared to those who tested negative for COVID-19 and were living with cirrhosis [14]. Several other studies conducted in the U.S and Europe revealed higher rates of hospitalization and mortality among COVID-19 patients with chronic liver diseases [4,6,15].

We found that several demographic characteristics, including older age; male sex; low income level; the presence of many comorbidities, including cirrhosis; vaccination status; and specific COVID-19 variants of concern were significant risk factors for hospitalization and ICU admission, which was consistent with evidence from elsewhere [16,17].

In this study, we also examined the potential impact of age in the associations of cirrhosis and hospitalization and ICU admission, considered separately, by stratifying our analysis by age-groups. After age stratification, we found that the highest magnitude of association for cirrhosis and these outcomes was observed in the youngest age-group (18–49 age-group). For example, among this age-group, the odds of being admitted to the ICU following COVID-19 were 581% higher for individuals with cirrhosis, compared to those without cirrhosis. Although to a lower degree, this is similar to the 198% (for the 50–69 age-group) and 117% (for the 70+ age-group) increased odds of being admitted to the ICU during the first 14 days of SARS-CoV-2 infection seen in individuals with cirrhosis compared to those without it. The odds of hospitalization among individuals with cirrhosis followed a similar trend, although to a lesser degree. This observation may be attributed to the higher likelihood of substance use and concurrent HCV infection among younger individuals, exposing them to additional risks. In contrast, older individuals who have survived and subsequently developed cirrhosis later in life might be benefitting from various factors. For instance, they may experience a less severe disease progression and maintain stronger connections with the healthcare system, facilitating the management of both cirrhosis and COVID-related conditions.

Our findings provide evidence on the association between health disparities and inequities and COVID-19 health outcomes. Income level was significantly associated with increased hospitalization and ICU admission; the odds of ICU admission decreased with increasing income level. Of all the income quintiles, the highest (fifth) quintile had the least magnitude of association with both hospitalization and ICU admission. The health authority regions where individuals lived were also associated with hospitalization and ICU admission. Indeed, there were 87% and 65% increases in the odds of hospitalization and ICU admission, respectively, for adults living in the Northern health region of the province (relative to the Vancouver Coastal health region). This may be related to the health inequities in access to care in remote/rural areas. These inequities should be taken into consideration by policy makers, health care professionals, researchers, or other stakeholders when developing health prevention and promotion programs.

The increased susceptibility of individuals with cirrhosis to severe COVID-19 outcomes can be attributed to two key factors. Firstly, cirrhosis-induced immune dysfunction renders these patients more prone to infections and related complications, thereby amplifying their vulnerability to severe outcomes of disease, including COVID-19. Secondly, this population is more inclined to engage in injection drug use, which serves as an additional risk factor, further elevating their likelihood of experiencing severe outcomes related to COVID-19.

Our study had several strengths and limitations. The large sample-size of our study is a major strength. Having used a population-based cohort, we are confident of the representativeness and generalizability of our findings. Also, we were able to assess information on income level, as well as regional differences, which helped to assess the role of social determinants of health in the association between cirrhosis and COVID-19 outcomes. Although we consider this to be an important strength, income level could only be measured at the geographic level and not at the individual level. Also, our use of administrative health data limited our ability to consider other variables that may be relevant in this investigation, particularly race/ethnicity-based data and other information related to social determinants of health. Assessment of ethnic/racial disparities would be important given the higher prevalence of hepatitis B and C virus infections, diabetes, and other risk factors among various racial groups. Additional limitations include potential residual confounding and our inability to infer causation given the observational nature of our study design.

## 5. Conclusions

Our assessment indicates that cirrhosis is associated with increased odds of hospitalization and ICU admission among COVID-19 patients. This not only underscores the importance of understanding the implication of cirrhosis in the context of COVID-19 but also highlights the value of population-level studies in informing public health strategies and interventions. Given the links between cirrhosis, immune dysfunction, and elevated risk of infections due to an immunocompromised status, assessing the risk of COVID-19 severe outcomes among individuals with cirrhosis provides evidence to support targeted interventions aimed at protecting individuals with cirrhosis from COVID-19-related outcomes. These may include prioritization for vaccination, health promotion messaging, preventive interventions, and population health assessment.

## Figures and Tables

**Figure 1 viruses-16-00351-f001:**
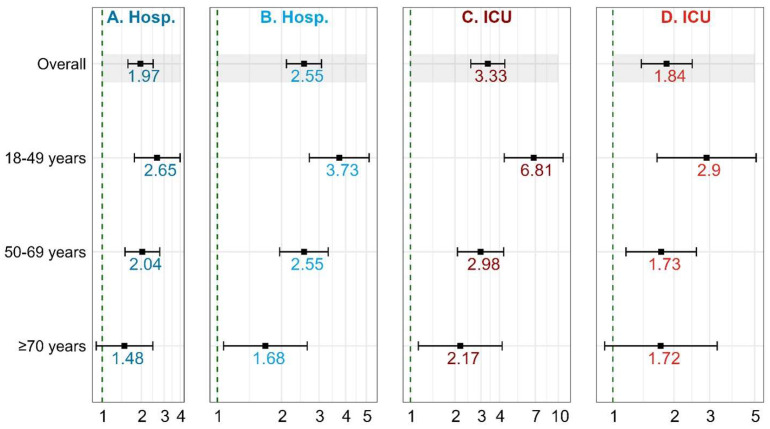
Association of cirrhosis with severe COVID-19-related outcomes (Hospitalization, ICU admission). (**A**) Outcome hospitalization—Hospitalization (minus ICU) vs. Non-hospitalization; (**B**) Outcome hospitalization—Hospitalization (includes ICU) vs. Non-hospitalization; (**C**) Outcome ICU—ICU vs. Non-hospitalization; (**D**) Outcome ICU—ICU vs. Hospitalization.

**Table 1 viruses-16-00351-t001:** Distribution of characteristics in confirmed COVID-19 adult cases during 2021 according to acute severity, BC COVID-19 Cohort.

	No Hospitalization	Hospital Admission	ICU Admission	Overall	*p*-Value
	(*N* = 153,963)	(*N* = 6035)	(*N* = 2511)	(*N* = 162,509)	
Sex					
Female	77,213 (50.2%)	2728 (45.2%)	928 (37.0%)	80,869 (49.8%)	<0.001
Male	76,750 (49.8%)	3307 (54.8%)	1583 (63.0%)	81,640 (50.2%)	
Age (years)					
Median (Q1–Q3)	37 (27–50)	60 (44–73)	60 (48–70)	38 (28–52)	<0.001
Age-group					
<20 Years	5852 (3.8%)	28 (0.5%)	13 (0.5%)	5893 (3.6%)	<0.001
20–29 Years	42,304 (27.5%)	437 (7.2%)	96 (3.8%)	42,837 (26.4%)	
30–39 Years	37,305 (24.2%)	758 (12.6%)	245 (9.8%)	38,308 (23.6%)	
40–49 Years	27,722 (18.0%)	793 (13.1%)	321 (12.8%)	28,836 (17.7%)	
50–59 Years	21,223 (13.8%)	988 (16.4%)	546 (21.7%)	22,757 (14.0%)	
60–69 Years	12,939 (8.4%)	1119 (18.5%)	620 (24.7%)	14,678 (9.0%)	
70–79 Years	4925 (3.2%)	951 (15.8%)	514 (20.5%)	6390 (3.9%)	
80+ Years	1693 (1.1%)	961 (15.9%)	156 (6.2%)	2810 (1.7%)	
Health authority					
Fraser	69,169 (44.9%)	2557 (42.4%)	985 (39.2%)	72,711 (44.7%)	<0.001
Interior	25,138 (16.3%)	1163 (19.3%)	540 (21.5%)	26,841 (16.5%)	
Northern	11,781 (7.7%)	788 (13.1%)	355 (14.1%)	12,924 (8.0%)	
Vancouver Coastal	34,369 (22.3%)	1105 (18.3%)	441 (17.6%)	35,915 (22.1%)	
Vancouver Island	12,283 (8.0%)	414 (6.9%)	189 (7.5%)	12,886 (7.9%)	
Unknown	1223 (0.8%)	8 (0.1%)	1 (0.0%)	1232 (0.8%)	
Income (Quintiles)					
1-Q	28,991 (18.8%)	1833 (30.4%)	746 (29.7%)	31,570 (19.4%)	<0.001
2-Q	29,034 (18.9%)	1243 (20.6%)	542 (21.6%)	30,819 (19.0%)	
3-Q	28,296 (18.4%)	1076 (17.8%)	427 (17.0%)	29,799 (18.3%)	
4-Q	28,433 (18.5%)	890 (14.7%)	395 (15.7%)	29,718 (18.3%)	
5-Q	25,721 (16.7%)	734 (12.2%)	287 (11.4%)	26,742 (16.5%)	
Missing/unknown	13,488 (8.8%)	259 (4.3%)	114 (4.5%)	13,861 (8.5%)	
Asthma					
No	133,442 (86.7%)	4932 (81.7%)	2046 (81.5%)	140,420 (86.4%)	<0.001
Yes	20,521 (13.3%)	1103 (18.3%)	465 (18.5%)	22,089 (13.6%)	
Cirrhosis					
No	153,425 (99.7%)	5903 (97.8%)	2413 (96.1%)	161,741 (99.5%)	<0.001
Yes	538 (0.3%)	132 (2.2%)	98 (3.9%)	768 (0.5%)	
Cancer, lymphoma					
No	153,257 (99.5%)	5920 (98.1%)	2454 (97.7%)	161,631 (99.5%)	<0.001
Yes	706 (0.5%)	115 (1.9%)	57 (2.3%)	878 (0.5%)	
Cancer, solid					
No	138,046 (89.7%)	4740 (78.5%)	2007 (79.9%)	144,793 (89.1%)	<0.001
Yes	15,917 (10.3%)	1295 (21.5%)	504 (20.1%)	17,716 (10.9%)	
Cancer, metastatic					
No	151,060 (98.1%)	5713 (94.7%)	2374 (94.5%)	159,147 (97.9%)	<0.001
Yes	2903 (1.9%)	322 (5.3%)	137 (5.5%)	3362 (2.1%)	
Chronic kidney disease					
No	149,733 (97.3%)	4928 (81.7%)	2013 (80.2%)	156,674 (96.4%)	<0.001
Yes	4230 (2.7%)	1107 (18.3%)	498 (19.8%)	5835 (3.6%)	
Chronic obstructive pulmonary disease					
No	151,856 (98.6%)	5426 (89.9%)	2244 (89.4%)	159,526 (98.2%)	<0.001
Yes	2107 (1.4%)	609 (10.1%)	267 (10.6%)	2983 (1.8%)	
Depression					
No	115,755 (75.2%)	3682 (61.0%)	1549 (61.7%)	120,986 (74.4%)	<0.001
Yes	38,208 (24.8%)	2353 (39.0%)	962 (38.3%)	41,523 (25.6%)	
Diabetes mellitus (DM, treatment)					
Non-DM	144,493 (93.8%)	4609 (76.4%)	1803 (71.8%)	150,905 (92.9%)	<0.001
DM, non-insulin-dependent	7817 (5.1%)	1025 (17.0%)	462 (18.4%)	9304 (5.7%)	
DM, insulin-dependent	1653 (1.1%)	401 (6.6%)	246 (9.8%)	2300 (1.4%)	
Obesity					
No	149,744 (97.3%)	5766 (95.5%)	2301 (91.6%)	157,811 (97.1%)	<0.001
Yes	4219 (2.7%)	269 (4.5%)	210 (8.4%)	4698 (2.9%)	
Weight loss					
No	151,680 (98.5%)	5706 (94.5%)	2413 (96.1%)	159,799 (98.3%)	<0.001
Yes	2283 (1.5%)	329 (5.5%)	98 (3.9%)	2710 (1.7%)	
Acute myocardial infarction					
No	152,839 (99.3%)	5743 (95.2%)	2404 (95.7%)	160,986 (99.1%)	<0.001
Yes	1124 (0.7%)	292 (4.8%)	107 (4.3%)	1523 (0.9%)	
Chronic heart disease *					
No	148,425 (96.4%)	4877 (80.8%)	2039 (81.2%)	155,341 (95.6%)	<0.001
Yes	5538 (3.6%)	1158 (19.2%)	472 (18.8%)	7168 (4.4%)	
Heart failure					
No	152,687 (99.2%)	5546 (91.9%)	2329 (92.8%)	160,562 (98.8%)	<0.001
Yes	1276 (0.8%)	489 (8.1%)	182 (7.2%)	1947 (1.2%)	
Hypertension					
No	136,548 (88.7%)	3621 (60.0%)	1495 (59.5%)	141,664 (87.2%)	<0.001
Yes	17,415 (11.3%)	2414 (40.0%)	1016 (40.5%)	20,845 (12.8%)	
Ischemic heart disease					
No	148,926 (96.7%)	5031 (83.4%)	2107 (83.9%)	156,064 (96.0%)	<0.001
Yes	5037 (3.3%)	1004 (16.6%)	404 (16.1%)	6445 (4.0%)	
Alcohol misuse					
No	145,687 (94.6%)	5165 (85.6%)	2181 (86.9%)	153,033 (94.2%)	<0.001
Yes	8276 (5.4%)	870 (14.4%)	330 (13.1%)	9476 (5.8%)	
Injection drug use					
No	145,635 (94.6%)	5165 (85.6%)	2202 (87.7%)	153,002 (94.1%)	<0.001
Yes	8328 (5.4%)	870 (14.4%)	309 (12.3%)	9507 (5.9%)	
Intellectual and developmental disabilities					
No	153,068 (99.4%)	5957 (98.7%)	2483 (98.9%)	161,508 (99.4%)	<0.001
Yes	895 (0.6%)	78 (1.3%)	28 (1.1%)	1001 (0.6%)	
Immunosuppression					
No	150,450 (97.7%)	5664 (93.9%)	2321 (92.4%)	158,435 (97.5%)	<0.001
Yes	3513 (2.3%)	371 (6.1%)	190 (7.6%)	4074 (2.5%)	
Alzheimer/dementia					
No	153,729 (99.8%)	5895 (97.7%)	2496 (99.4%)	162,120 (99.8%)	<0.001
Yes	234 (0.2%)	140 (2.3%)	15 (0.6%)	389 (0.2%)	
Epilepsy					
No	152,777 (99.2%)	5919 (98.1%)	2467 (98.2%)	161,163 (99.2%)	<0.001
Yes	1186 (0.8%)	116 (1.9%)	44 (1.8%)	1346 (0.8%)	
Parkinsonism					
No	153,878 (99.9%)	5990 (99.3%)	2505 (99.8%)	162,373 (99.9%)	<0.001
Yes	85 (0.1%)	45 (0.7%)	6 (0.2%)	136 (0.1%)	
Rheumatoid arthritis					
No	152,497 (99.0%)	5839 (96.8%)	2434 (96.9%)	160,770 (98.9%)	<0.001
Yes	1466 (1.0%)	196 (3.2%)	77 (3.1%)	1739 (1.1%)	
Schizophrenia and psychotic disorders					
No	151,813 (98.6%)	5656 (93.7%)	2424 (96.5%)	159,893 (98.4%)	<0.001
Yes	2150 (1.4%)	379 (6.3%)	87 (3.5%)	2616 (1.6%)	
Variants of concern					
Non-VOC	9353 (6.1%)	398 (6.6%)	158 (6.3%)	9909 (6.1%)	<0.001
Delta	38,540 (25.0%)	2191 (36.3%)	989 (39.4%)	41,720 (25.7%)	
Alpha	17,270 (11.2%)	713 (11.8%)	290 (11.5%)	18,273 (11.2%)	
Beta	98 (0.1%)	4 (0.1%)	3 (0.1%)	105 (0.1%)	
Gamma	12,827 (8.3%)	727 (12.0%)	356 (14.2%)	13,910 (8.6%)	
Not sequenced	68,917 (44.8%)	1916 (31.7%)	701 (27.9%)	71,534 (44.0%)	
Omicron	6958 (4.5%)	86 (1.4%)	14 (0.6%)	7058 (4.3%)	
Vaccination status					
Not vaccinated	90,252 (58.6%)	4439 (73.6%)	2069 (82.4%)	96,760 (59.5%)	<0.001
Partially vaccinated	16,063 (10.4%)	867 (14.4%)	269 (10.7%)	17,199 (10.6%)	
Vaccinated	47,648 (30.9%)	729 (12.1%)	173 (6.9%)	48,550 (29.9%)	

* Combination of acute myocardial infarction, heart failure, and ischemic heart disease.

**Table 2 viruses-16-00351-t002:** Distribution of characteristics in confirmed COVID-19 adult cases during 2021 according to cirrhosis status, BC COVID-19 Cohort.

	No Cirrhosis	Cirrhosis	Overall	*p*-Value
	(*N* = 161,741)	(*N* = 768)	(*N* = 162,509)	
Severity (acute)				
No hospitalization	153,425 (94.9%)	538 (70.1%)	153,963 (94.7%)	<0.001
Hospitalization	5903 (3.6%)	132 (17.2%)	6035 (3.7%)	
ICU admission	2413 (1.5%)	98 (12.8%)	2511 (1.5%)	
Sex				
Female	80,501 (49.8%)	368 (47.9%)	80,869 (49.8%)	0.591
Male	81,240 (50.2%)	400 (52.1%)	81,640 (50.2%)	
Age (years)				
Median (Q1–Q3)	38 (28–52)	54 (43–64)	38 (28–52)	<0.001
Age-group				
<20 Years	5890 (3.6%)	3 (0.4%)	5893 (3.6%)	<0.001
20–29 Years	42,805 (26.5%)	32 (4.2%)	42,837 (26.4%)	
30–39 Years	38,204 (23.6%)	104 (13.5%)	38,308 (23.6%)	
40–49 Years	28,672 (17.7%)	164 (21.4%)	28,836 (17.7%)	
50–59 Years	22,574 (14.0%)	183 (23.8%)	22,757 (14.0%)	
60–69 Years	14,497 (9.0%)	181 (23.6%)	14,678 (9.0%)	
70–79 Years	6312 (3.9%)	78 (10.2%)	6390 (3.9%)	
80+ Years	2787 (1.7%)	23 (3.0%)	2810 (1.7%)	
Health authority				
Fraser	72,407 (44.8%)	304 (39.6%)	72,711 (44.7%)	<0.001
Interior	26,690 (16.5%)	151 (19.7%)	26,841 (16.5%)	
Northern	12,785 (7.9%)	139 (18.1%)	12,924 (8.0%)	
Vancouver Coastal	35,804 (22.1%)	111 (14.5%)	35,915 (22.1%)	
Vancouver Island	12,823 (7.9%)	63 (8.2%)	12,886 (7.9%)	
Unknown	1232 (0.8%)	0 (0%)	1232 (0.8%)	
Income (Quintiles)				
1-Q	31,302 (19.4%)	268 (34.9%)	31,570 (19.4%)	<0.001
2-Q	30,661 (19.0%)	158 (20.6%)	30,819 (19.0%)	
3-Q	29,673 (18.3%)	126 (16.4%)	29,799 (18.3%)	
4-Q	29,587 (18.3%)	131 (17.1%)	29,718 (18.3%)	
5-Q	26,663 (16.5%)	79 (10.3%)	26,742 (16.5%)	
Missing/unknown	13,855 (8.6%)	6 (0.8%)	13,861 (8.5%)	
Asthma				
No	139,808 (86.4%)	612 (79.7%)	140,420 (86.4%)	<0.001
Yes	21,933 (13.6%)	156 (20.3%)	22,089 (13.6%)	
Cancer, lymphoma				
No	160,881 (99.5%)	750 (97.7%)	161,631 (99.5%)	<0.001
Yes	860 (0.5%)	18 (2.3%)	878 (0.5%)	
Cancer, solid				
No	144,256 (89.2%)	537 (69.9%)	144,793 (89.1%)	<0.001
Yes	17,485 (10.8%)	231 (30.1%)	17,716 (10.9%)	
Cancer, metastatic				
No	158,462 (98.0%)	685 (89.2%)	159,147 (97.9%)	<0.001
Yes	3279 (2.0%)	83 (10.8%)	3362 (2.1%)	
Chronic kidney disease				
No	156,080 (96.5%)	594 (77.3%)	156,674 (96.4%)	<0.001
Yes	5661 (3.5%)	174 (22.7%)	5835 (3.6%)	
Chronic obstructive pulmonary disease				
No	158,822 (98.2%)	704 (91.7%)	159,526 (98.2%)	<0.001
Yes	2919 (1.8%)	64 (8.3%)	2983 (1.8%)	
Depression				
No	120,627 (74.6%)	359 (46.7%)	120,986 (74.4%)	<0.001
Yes	41,114 (25.4%)	409 (53.3%)	41,523 (25.6%)	
Diabetes mellitus (DM, treatment)				
Non-DM	150,323 (92.9%)	582 (75.8%)	150,905 (92.9%)	<0.001
DM, non-insulin-dependent	9176 (5.7%)	128 (16.7%)	9304 (5.7%)	
DM, insulin-dependent	2242 (1.4%)	58 (7.6%)	2300 (1.4%)	
Obesity				
No	157,091 (97.1%)	720 (93.8%)	157,811 (97.1%)	<0.001
Yes	4650 (2.9%)	48 (6.3%)	4698 (2.9%)	
Weight loss				
No	159,091 (98.4%)	708 (92.2%)	159,799 (98.3%)	<0.001
Yes	2650 (1.6%)	60 (7.8%)	2710 (1.7%)	
Acute myocardial infarction				
No	160,240 (99.1%)	746 (97.1%)	160,986 (99.1%)	<0.001
Yes	1501 (0.9%)	22 (2.9%)	1523 (0.9%)	
Chronic heart disease *				
No	154,710 (95.7%)	631 (82.2%)	155,341 (95.6%)	<0.001
Yes	7031 (4.3%)	137 (17.8%)	7168 (4.4%)	
Heart failure				
No	159,865 (98.8%)	697 (90.8%)	160,562 (98.8%)	<0.001
Yes	1876 (1.2%)	71 (9.2%)	1947 (1.2%)	
Hypertension				
No	141,173 (87.3%)	491 (63.9%)	141,664 (87.2%)	<0.001
Yes	20,568 (12.7%)	277 (36.1%)	20,845 (12.8%)	
Ischemic heart disease				
No	155,398 (96.1%)	666 (86.7%)	156,064 (96.0%)	<0.001
Yes	6343 (3.9%)	102 (13.3%)	6445 (4.0%)	
Alcohol misuse				
No	152,590 (94.3%)	443 (57.7%)	153,033 (94.2%)	<0.001
Yes	9151 (5.7%)	325 (42.3%)	9476 (5.8%)	
Injection drug use				
No	152,471 (94.3%)	531 (69.1%)	153,002 (94.1%)	<0.001
Yes	9270 (5.7%)	237 (30.9%)	9507 (5.9%)	
Intellectual and developmental disabilities				
No	160,746 (99.4%)	762 (99.2%)	161,508 (99.4%)	0.842
Yes	995 (0.6%)	6 (0.8%)	1001 (0.6%)	
Immunosuppression				
No	157,757 (97.5%)	678 (88.3%)	158,435 (97.5%)	<0.001
Yes	3984 (2.5%)	90 (11.7%)	4074 (2.5%)	
Alzheimer/dementia				
No	161,360 (99.8%)	760 (99.0%)	162,120 (99.8%)	<0.001
Yes	381 (0.2%)	8 (1.0%)	389 (0.2%)	
Epilepsy				
No	160,422 (99.2%)	741 (96.5%)	161,163 (99.2%)	<0.001
Yes	1319 (0.8%)	27 (3.5%)	1346 (0.8%)	
Parkinsonism				
No	161,606 (99.9%)	767 (99.9%)	162,373 (99.9%)	0.905
Yes	135 (0.1%)	1 (0.1%)	136 (0.1%)	
Rheumatoid arthritis				
No	160,023 (98.9%)	747 (97.3%)	160,770 (98.9%)	<0.001
Yes	1718 (1.1%)	21 (2.7%)	1739 (1.1%)	
Schizophrenia and psychotic disorders				
No	159,164 (98.4%)	729 (94.9%)	159,893 (98.4%)	<0.001
Yes	2577 (1.6%)	39 (5.1%)	2616 (1.6%)	
Variants of concern				
Non-VOC	9822 (6.1%)	87 (11.3%)	9909 (6.1%)	<0.001
Delta	41,491 (25.7%)	229 (29.8%)	41,720 (25.7%)	
Alpha	18,206 (11.3%)	67 (8.7%)	18,273 (11.2%)	
Beta	101 (0.1%)	4 (0.5%)	105 (0.1%)	
Gamma	13,871 (8.6%)	39 (5.1%)	13,910 (8.6%)	
Not sequenced	71,206 (44.0%)	328 (42.7%)	71,534 (44.0%)	
Omicron	7044 (4.4%)	14 (1.8%)	7058 (4.3%)	
Vaccination status				
Not vaccinated	96,317 (59.6%)	443 (57.7%)	96,760 (59.5%)	0.00111
Partially vaccinated	17,082 (10.6%)	117 (15.2%)	17,199 (10.6%)	
Vaccinated	48,342 (29.9%)	208 (27.1%)	48,550 (29.9%)	

* Combination of acute myocardial infarction, heart failure, and ischemic heart disease.

**Table 3 viruses-16-00351-t003:** Association of cirrhosis with severe COVID-19-related outcomes (Hospitalization, ICU admission) in multivariable logistic regression. * analysis among confirmed adult cases during 2021, BC COVID-19 Cohort, stratified by age-group.

	Hospitalization				ICU Admission			
	AOR	LCI	UCI	*p*-Value	AOR	LCI	UCI	*p*-Value
Overall	1.97	1.58	2.47	<0.001	3.33	2.56	4.35	<0.001
Ages 18–49	2.65 ⁰†	1.77	3.99	<0.001	6.81 ‡	4.31	10.78	<0.001
Ages 50–69	2.04 ⁰	1.50	2.78	<0.001	2.98	2.08	4.27	<0.001
Ages ≥ 70	1.48	0.90	2.46	0.125	2.17 †	1.13	4.17	0.02

AOR: adjusted odds ratio; LCI: lower confidence interval; UCI: upper confidence interval. * Models adjusted for sex, age-group (where applicable), asthma, chronic obstructive pulmonary disease, chronic kidney disease, diabetes mellitus, heart failure, hypertension, injection drug use, alcohol misuse, immunosuppression, Alzheimer/dementia, schizophrenia and psychotic disorders, multiple sclerosis, parkinsonism, rheumatoid arthritis, obesity, weight loss, intellectual and developmental disabilities, cancer (lymphoma), cancer (metastatic), variant of concern, vaccination status, income and health authority. ⁰ VOC: Alpha and Beta were collapsed together due to low counts. † Parkinsonism was removed from model due to zero counts. ‡ Alzheimer/dementia and Parkinsonism were removed from model due to zero counts.

## Data Availability

The study is based on data contained in various provincial registries and databases. Access to data could be requested through the BC Centre for Disease Control Institutional Data Access by researchers who meet the criteria for access to confidential data. Requests for the data may be sent to datarequest@bccdc.ca.

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
