# Peer review of "Risk of Severe COVID-19-Related Outcomes among Patients with Cirrhosis: A Population-Based Cohort Study in Canada"

_viruses, 2024, doi:10.3390/v16030351_

Round 1

Reviewer 1 Report

Comments and Suggestions for Authors

There have been studies suggesting that individuals with pre-existing liver conditions, including cirrhosis, may be at an increased risk of more severe illness from COVID-19. Additionally, the association between liver disease, age, and admission to intensive care unit (ICU) during COVID-19 can be influenced by different factors, including the specific characteristics of the individuals, compromised immune function, underlying health conditions, liver injury, and the stage of cirrhosis.

This is an interesting piece of work in an extremely important area, which raises questions that are useful for prospective studies. 

The authors use logistic regression models to evaluate odds ratios (ORs), which provide a measure of the association between the presence of liver cirrhosis and the odds of being hospitalized due to COVID-19. While multivariable logistic regression models are very valuable tools in epidemiological studies, there are some potential limitations to consider when using them to evaluate the associations between stages of liver cirrhosis and COVID-19-related hospitalization. Some of these limitations include e.g. observational nature, residual confounding, data quality, caution v. association, heterogeneity, and selection bias. It´s important to note the association between cirrhosis and hospitalization is very complex and multifactorial.

I have only minor comments that needed to be addressed:

KEYWORDSCOVID-19; cirrhosis; risk; hospitalization; risk factors; Canada. (The title words should not be repeated in Keywords”).

 In Discussion/Conclusions  I would suggest that the author  compare the results they obtained with other studies. Please highlight the limitations and implications for researchers carrying out future studies in the same field, and for public health practice.

Comments on the Quality of English Language

     The quality of English language is ok

Author Response

Reviewer 1

There have been studies suggesting that individuals with pre-existing liver conditions, including cirrhosis, may be at an increased risk of more severe illness from COVID-19. Additionally, the association between liver disease, age, and admission to intensive care unit (ICU) during COVID-19 can be influenced by different factors, including the specific characteristics of the individuals, compromised immune function, underlying health conditions, liver injury, and the stage of cirrhosis.

This is an interesting piece of work in an extremely important area, which raises questions that are useful for prospective studies. 

The authors use logistic regression models to evaluate odds ratios (ORs), which provide a measure of the association between the presence of liver cirrhosis and the odds of being hospitalized due to COVID-19. While multivariable logistic regression models are very valuable tools in epidemiological studies, there are some potential limitations to consider when using them to evaluate the associations between stages of liver cirrhosis and COVID-19-related hospitalization. Some of these limitations include e.g. observational nature, residual confounding, data quality, caution v. association, heterogeneity, and selection bias. It´s important to note the association between cirrhosis and hospitalization is very complex and multifactorial.

RESPONSE: We appreciate your insightful comments.

I have only minor comments that needed to be addressed:

KEYWORDS: COVID-19; cirrhosis; risk; hospitalization; risk factors; Canada. (The title words should not be repeated in” Keywords”).

RESPONSE: We appreciate your suggestion and agree with this rule of thumb. However, we also acknowledge that circumstances may arise wherein the overlapping of title words and keywords remains imperative, particularly when these words are quite important. In instances where these words hold substantial relevance and the journal lacks explicit prohibitions, such as in our present scenario, we believe that their inclusion should be deemed permissible. Our assessment suggests that the journal "Viruses" does not impose restrictions on this matter, as evidenced by several recently published studies: https://www.mdpi.com/1999-4915/16/2/285; https://www.mdpi.com/1999-4915/16/2/281

In Discussion/Conclusions I would suggest that the author compare the results they obtained with other studies.

RESPONSE: In addition to what was already presented, we have reported findings from other studies whose findings compare with ours. This is included in the Discussion section.

Please highlight the limitations and implications for researchers carrying out future studies in the same field, and for public health practice.

RESPONSE: We have now incorporated further limitations beyond those previously outlined.

Reviewer 2 Report

Comments and Suggestions for Authors

1.       Why did the authors report this study so late? Please discuss and justify.

2.       Is there any impact of vaccination on this study? Did the authors follow the vaccination status during this study? Please discuss and justify.

3.       What was the time gap between SARS-CoV-2 infection and cirrhosis reported in patients?

4.       Is there any sex bias reported in this study? Please discuss.

5.       Did the authors follow up on the drugs used by the patients during the study and their effect on the liver? Most of the drugs have some effect on the liver if used for a longer time. Please discuss and justify.

6.       There are limitations associated with this study. Please include a separate section describing the limitations of the current study.

Author Response

Reviewer 2

  1. Why did the authors report this study so late? Please discuss and justify.

RESPONSE: This study uses a large integrated dataset which required extensive data preparation, linkage, and analysis. Ensuring data accuracy, reliability and quality takes time. Our research team prioritizes ensuring the accuracy and reliability of data through rigorous quality assurance procedures, including thorough validation and verification processes.

  1. Is there any impact of vaccination on this study? Did the authors follow the vaccination status during this study? Please discuss and justify.

RESPONSE: Yes, we assessed vaccination status of participants and this data was included in our analyses. Please see study results. Given the focus of this study was not assessment of vaccination impact, we did not highlight the impact of vaccination although we have provided the results. You will notice that in the analyses, vaccination was associated with reduced COVID-19 severe outcomes. We have published other status that investigated this. Velásquez García et al., 2023 [https://pubmed.ncbi.nlm.nih.gov/36503044/#:~:text=Among%20vaccinated%20individuals%2C%20an%20increased,%2C%2017.71)%2C%20and%20%E2%89%A580]

  1. What was the time gap between SARS-CoV-2 infection and cirrhosis reported in patients?

Given that the identification of cirrhosis is made through the analysis of administrative data with the help of a validated algorithm that does not take into account the length of the disease, we can determine with precision only the presence of the condition at the moment when the SARS-CoV-2 sample was collected.

  1. Is there any sex bias reported in this study? Please discuss.

RESPONSE: We adjusted for sex in our analyses. Please see Table 3 footnotes as well as the supplementary analyses.

  1. Did the authors follow up on the drugs used by the patients during the study and their effect on the liver? Most of the drugs have some effect on the liver if used for a longer time. Please discuss and justify.

RESPONSE: Thank you for the observation. No, we did not take into account prescription medications, however, we did adjust for a wide variety of conditions and comorbidities. By doing so, implicit adjustment for prescribed medications is present to some degree. Furthermore, as drug-related variables would be highly correlated to comorbidities, their inclusion in regression models (in this scenario, more than 20 adjustment variables) could generate multicollinearity issues.

  1. There are limitations associated with this study. Please include a separate section describing the limitations of the current study.

RESPONSE: Thank you for the suggestion. We have included additional limitations.

Reviewer 3 Report

Comments and Suggestions for Authors

The authors tried to study relationships between COVID-19 severity and cirrhosis based on a population-based cohort study in Canada. An aim of the study was fine, while data analyses and presentations were not.

Major comments

Between the title of the study and contents were incompatible. If the authors aim to study between covid-19 severity and cirrhosis, numerous unnecessary data were analyzed and presented. Moreover, all tables were nearly raw data. These should be improved and sophisticated.

Minor comments

The authors should carefully describe everything including abbreviation usage priority, space, etc according to the appropriate formats.    

Delete Lines 109-111, Line 166.

Author Response

Reviewer 3

The authors tried to study relationships between COVID-19 severity and cirrhosis based on a population-based cohort study in Canada. An aim of the study was fine, while data analyses and presentations were not.

Major comments

Between the title of the study and contents were incompatible. If the authors aim to study between covid-19 severity and cirrhosis, numerous unnecessary data were analyzed and presented. Moreover, all tables were nearly raw data. These should be improved and sophisticated.

RESPONSE: We acknowledge the possibility that the number of variables employed in our study may be perceived as extensive. However, this deliberate choice was made to enhance the methodological rigor of our analysis, with a specific focus on mitigating the potential for residual confounding.

Minor comments

The authors should carefully describe everything including abbreviation usage priority, space, etc according to the appropriate formats. 

RESPONSE: We have ensured that all abbreviations and terminologies are well described and explained.

Delete Lines 109-111, Line 166.

RESPONSE: Thank you. We have deleted this section.

Round 2

Reviewer 2 Report

Comments and Suggestions for Authors

The authors successfully responded to the reviewer's comments and updated the manuscript as well. 

Reviewer 3 Report

Comments and Suggestions for Authors

Although the revised manuscript still contained little relevant data, I felt that this work was valuable for publication in Virology.